# Tubeimoside-1 Inhibits Glioblastoma Growth, Migration, and Invasion via Inducing Ubiquitylation of MET

**DOI:** 10.3390/cells8080774

**Published:** 2019-07-25

**Authors:** Jiangjun Cao, Erhu Zhao, Qingzong Zhu, Juanli Ji, Zekun Wei, Bo Xu, Hongjuan Cui

**Affiliations:** 1Chongqing Engineering Research Center of Antitumor Natural Drugs, Chongqing Three Gorges Medical College, Chongqing 404120, China; 2State Key Laboratory of Silkworm Genome Biology, Southwest University, Chongqing 400715, China; 3Institute of Medicine of Southwest University, Southwest University, Chongqing 400715, China

**Keywords:** glioblastoma, tubeimoside-1, MET

## Abstract

Tubeimoside-1 (TBMS1) is one of the extracts of rhizoma bolbostemmae, which has remarkable anti-cancer function in the treatment of esophagus and gastric cancer in traditional Chinese medicine. However the mechanisms of its anti-cancer function is remain unclear. In this study, we demonstrate that TBMS1 could inhibit cell growth and metastasis in glioblastoma. MET is a member of the receptor tyrosine kinase family, which amplifies frequently in various human cancers. As an important proto-oncogene, multiple inhibitors have been developed for the therapy of cancers. Here, we found TBMS1 could reduce/decrease the protein level of MET via increasing its Ubiquitination degradation. Therefore, TBMS1 is a promising compound for the treatment of glioblastoma and an inhibitor of MET.

## 1. Introduction

Rhizoma bolbostemmae is a kind of Chinese traditional medicinal materials, which is used therapeutically for Mammary carbuncles, scrofula, and phlegm nodes. In Sichuan and Shanxi, rhizoma bolbostemae is regarded as an effective traditional Chinese herb for the treatment of esophagus and gastric cancer. Tubeimoside-1 (TBMS1) is the Chinese medicine monomer with a high yield (1.9%), water solubility, and stability extracted from the anti-cancer active ingredient saponin. In recent years, the underlying mechanisms of its anti-cancer function have been gradually revealed among multiple cancers. However, there are few studies in glioblastoma.

Glioblastoma (GBM) is the most common subtype tumor among malignant gliomas. As a malignant tumor with high metastasis and recurrence, the prognoses of radiotherapy and chemotherapy are poor [1,2]. However, with the emergence of important target genes, targeting therapy has become a potential approach for the treatment of glioblastoma. For instances, Crizotinib, an ATP-competitive small molecule inhibitor of MET, and Onartuzumab, a monovalent monoclonal antibody that binds to the extracellular domain of the MET receptor, have achieved certain results in clinical treatment [3]. However, the alterations of critical proteins including mTOR, FGFR1, EGFR, STAT3, and COX-2, which renders these agents ineffective, still blocks better results [4]. For this situation, finding more targeted drugs and clarifying their mechanisms will be an effective way to improve the therapeutic effect.

## 2. Materials and Methods

### 2.1. Cell Culture

Human glioblastoma cell lines (U87 and LN229) and human embryonic renal cell lines (293FT) were obtained from American Type Culture Collection (ATCC, Manassas, VA, USA), which were tested mycoplasma-negative. U87 and LN229 cells were cultured in Dulbecco’s modified Eagle’s medium (DMEM, Gibco, New York, NY, USA) added 10% fetal bovine serum (FBS, Gibco), 1% penicillin and streptomycin (P/S, Invitrogen, Carlsbad, CA, USA). 293FT cells were maintaining with DMEM added 1% G418 (Invitrogen), 2% glutamine (Invitrogen), 1% non-essential amino acids (Invitrogen), and 1% sodium pyruvate (Invitrogen). All experimentally-related cell lines were cultured under standard condition (5% CO_2_ at 37 °C).

### 2.2. Tubeimoside-1 Treatment

Tubeimoside-1 (molecular formula: C_63_H_98_O_29_, relative molecular mass: 1319.43) was obtained from Chengdu Must Biotechnology Company and dissolved in DMSO (20 mM) as stock solution. Glioblastoma cell lines, U87 and LN229 then was treated with tubeimoside-1 in different concentration (1, 5, 10, and 20 μM), DMSO was added as control. Microscopy (Olympus, Japan) was used to detect the morphology treated with different concentration (1, 5, 10, and 20 μM); simultaneously, hemocytometer was used for cell numbers counting. Every experiment was performed in triplicates independently.

### 2.3. Cell Viability Assay

The cell viability detection was performed by MTT (Sigma Aldrich, St. Louis, MO, USA) assay. We counted U87 and LN229 cells (800 cells per well) in logarithmic phase and then seeded in 96-well plates with medium (200 μL) and attached overnight. U87 and LN229 cells treated with tubeimoside-1 in different concentrations (1, 5, 10, and 20 μM). The control group treated with DMSO. At the specified time, cells were incubated with MTT (5 mg/mL, 20 μL per well) for 2 h in 37 °C incubator. DMSO (200 μL) was then used to solve the formazan. Microplate reader (Thermo Fisher, Waltham, MA, USA) was used to monitor the absorbance at 560 nm. All data above were analyzed by Graphpad. Every experiment was performed in triplicates independently.

### 2.4. BrdU Staining

BrdU staining was used to monitor cell proliferation. The 2 × 10^4^ of U87 and LN229 cells were seeded in 24-well plates respectively and attached overnight in incubator. Medium with different concentrations of Tubeimoside-1 (5, 10 μM) was then added and changed to glioblastoma cells. The control group treated with DMSO. After 48 h culture, 10 μg/mL BrdU (Sigma Aldrich, USA) was added into medium and incubated for 2 h. Finally, fixed with 4% paraformaldehyde in medium for 15 min. Then cells treated with 2 M HCL, 0.3% TritonX-100, and then blocking with 10% goat serum (ZSGB-Bio, Beijing, China). After blocking, Primary antibody (Abcam, Cambridge, MA, USA) and secondary antibodies (Abcam) are sequentially incubated with cells. Cells were stained with Hochest (Life, New York, NY, USA) before microscopic observation. BrdU-positive cells were counted randomly.

### 2.5. Flow Cytometry Analysis

Medium containing 5 μM tubeimoside-1 and DMSO was used to culture glioblastoma cells and then gathered for flow cytometry analysis at the scheduled time. For cell cycle assay, cells cultured with tubeimoside-1 and DMSO were gathered at 24 h and then washed with cold phosphate-buffered saline (PBS). After washing with PBS, cells fixed with 75% ethanol for 48 h at 4 °C. Washing thrice with PBS, cells were incubated in 150 μL PBS containing 1 μL RNaseA (Sigma Aldrich, USA) and 1 μL PI (BD, San Jose, CA, USA) for 30 min at 37 °C. BD accuri C6 flow cytometry (BD, San Jose, CA USA) was used to analysis cells. Cell cycle of glioblastoma cells were analyzed by Flow cytometry and FlowJo software. Every experiment was performed in triplicates independently.

### 2.6. Western Blot Analysis

RIPA lysis buffer (Beyotime, Shanghai, China) containing Phenylmethanesulfonyl fluoride (Beyotime) was used to lyse cells. After lysis, cell lysates were denatured for 30 min at 100 °C. Then, proteins were separated with 8% and 10% SDS-PAGE gel. Separated proteins were transferred to polyvinylidene difluoride membranes. The membranes blocked in 5% BSA (bovine serum albumin) at room temperature for 2 h. The primary antibodies against CDK1, Cyclin A2, Cyclin B1, MMP-7, MMP-2, Snail, Slug, p-MEK, P-ERK, AKT, p-AKT, and MET, which were purchased from Cell Signaling Technology (Proteintech, Chicago, IL, USA) as well as Tubulin purchased from Beyotime, incubated with membranes overnight at 4 °C. Membranes were then incubated with horseradish peroxidase-conjugated secondary antibody (HRP-conjugated secondary antibodies (goat anti-mouse IgG and goat anti-rabbit IgG, 1:10000, Beyotime, China) at room temperature for 2 h. Finally, ECL system (Beyotime, China) and ProXima chemiluminescence gel imaging system (Isogen, De Meern, Utrecht, The Netherlands) were used to visualize and captured proteins.

### 2.7. Transfection and Infection

For efficient transfection, 293FT cells (90% density) and plasmids including pCDH-CMV-MCS-EF1-copGFP-MET (Vector encoding human MET was obtained from YouBio, Changsha, China). The pCDH-CMV-MCS-EF1-copGFP vector was purchased from Addgene (Beijing, China) and empty vector were transfected into 293FT cells respectively by using ViaFect transfection reagent (Promega, Madison, WI, USA) according to the manufacturer’s operation manual. After 48 h of transfection, viral supernatants were collected and then infected into the glioblastoma cells helping with Polybrene. Puromycin was used for the selection of cells.

### 2.8. qRT-PCR Assay

Treated with DMSO or tubeimoside-1 for 1 day at 37 °C, cells’ total RNA was extracted with Trizol reagent and the experiment is according to the manufacturer’s instructions. Quantitative real-time PCR (qPCR) was used to detect the relative expression of different genes. The inverse transcription to cDNA of total RNA was performed by using M-MLV reverse transcriptase (Promega). And, GAPDH was used as a control. All mRNA relative expression levels were calculated utilizing the 2^−ΔΔCT^ method.

### 2.9. Soft Agar Assay

The colony formation ability of glioblastoma cells was detected by Soft agar assay. Base agar (1.5 mL per well) was consist of 2× DMEM (Gibco) and 0.6% agarose (Sigma-Aldrich, St. Louis, MO, USA). Medium containing Eight hundred cells, 0.3% agar and different concentrations tubeimoside-1 (5, 10 μM) were added onto base agar. After culturing for 21 days at 37 °C incubator, microscopy was used to capture colonies as well as MTT was used for count after staining.

### 2.10. Tumor Xenografts

The animal experiments have been approved by the animal ethics committee of Southwest University, and humanized experiments were conducted in accordance with the requirements of the Guidelines for the Care and Use of Laboratory Animals (Ministry of Science and Technology of China, 2006). Ethical Approval Number : IACUC-20190122-18. One-month-old female nude mice were fed in an SPF room. Then 100 μL basic medium with 1 × 10^6^ U87 cells were injected to subcutaneously on both sides. The mice were divided into two groups (6 mice per group) randomly after the injection for 1 week. At a later time, one group injected tubeimoside-1 (5 mg/kg) every two days and the other injected DMSO as a control. At the same time, tumor volume (tumor volume = (π/6) × length × width2) and the body weight were measured.

### 2.11. Orthotopic Implantation Assay

In orthotopic implantation assays, glioma U87 cells (1 × 10^5^ cells) were intracranially injected slowly into the brains of NOD/SCID mice (2 mm lateral and 1 mm anterior to the bregma, 3.5 mm deep). After one week, the mice were divided into two groups (6 mice per group) randomly and intraperitoneally injected with Tubeimoside-1 (5 mg/kg) or an equivalent amount of DMSO once daily for 15 days. All mice were euthanatized on the second day after the last administration. Brains were collected, fixed in neutral buffered formalin, and embedded in paraffin. Hematoxylin and eosin (H&E) staining were performed for evaluations of the tissues.

### 2.12. Ubiquitination Assay

Glioblastoma cells, treated with tubeimoside-1 and DMSO (10 μM) respectively, were incubated with MG132 (50 μg/mL, a proteasome inhibitor, purchased from Selleck, Houston, TX, USA) in 37 °C incubator for 6 h and then lysed with IP lysis buffer. MET antibody was incubated with lysate overnight and then incubated with IgG (Rabbit) to pull down the MET proteins. After five times washing with PBS, the protein denaturized for 30 min at 100 °C and then 8% SDS-PAGE gel was used to separation. Ubiquitination antibody was used to check the ubiquitination of MET.

### 2.13. Statistics Analysis

All statistics analyses were performed by Graphpad (GraphPad Prism 6). FlowJo (FlowJo 7.6) was used to analyze the cell cycle. Quantitative data were presented as mean ± S.D. (standard deviation). Significant different computation come from student’s *t*-test (* *p* < 0.05, ** *p* < 0.01, *** *p* < 0.001, *p*-value < 0.05 were considered as statistically significant).

## 3. Results

### 3.1. TBMS1 Inhibits Glioblastoma Cell Proliferation

Firstly, we investigated the IC50 of TBMS1 on glioblastoma cells, LN229 and U87 and normal nerve cells HEB. We found the medial lethal concentration of HEB is significantly higher than LN229 and U87 (Appendix A). Then, we examined the effects of TBMS1 on glioblastoma cells, LN229 and U87 treated with different concentrations of TBMS1 (1, 5, 10, and 20 μM, dimethyl sulfoxide (DMSO) was used as control) for 24 h. Observing by microscopy, LN229 and U87 cells treated with TBMS1 showed cell numbers decreased in a dose-dependent manner (Figure 1a,b). The assays 3-(4,5-dimethylthiazol-2-yl)-2,5-diphenyltetrazolium bromide (MTT), and bromodeoxyuridine (BrdU) were used to analyze cell growth and proliferation. MTT assay showed that glioblastoma cells treated with 1, 5, 10, and 20 μM TBMS1 showed sharp declines in the growth curve comparing to DMSO group (Figure 1c). From the above data, we will use 5 and 10 μm TBMS1 for subsequent experiments. Together, we found TBMS1 could inhibit the growth of glioblastoma cells.

### 3.2. TBMS1 Causes Glioblastoma Cells Cell Cycle Arrest at G2/M Phase

To figure out whether TBMS1 could cause cell cycle arrest, we preformed BrdU assay. We found that DNA synthesis reduced in the group treated with 5 and 10 μM TBMS1 compared to treat with DMSO (Figure 2a,b). Subsequently, we utilized flow cytometry analysis. The results showed the percentages of G_2_ phase of in both U87 and LN229 cells were significantly increased in a dose-dependent manner (Figure 2c,d). We further certify these results by detecting the expression of Cyclin A2, Cyclin B1, and CDK1 proteins which are essential for G_2_/M phase transition. As results showed, with the increasing of concentrations of TBMS1, the expression of Cyclin A2, Cyclin B1, and CDK1 proteins were highly inhibited in a dose-dependent manner (Figure 2e). Therefore, from the data above we found that TBMS1 blocks glioblastoma cells in G_2_/M phase by inhibiting the expression of Cyclin A2, Cyclin B1, and CDK1.

### 3.3. TBMS1 Blocks Glioblastoma Cells Migration and Invasion

Due to the high metastatic feature of glioblastoma, we investigated whether TBMS1 inhibits cell migration and invasion. Wound-healing assay were performed under different TBMS1 concentrations. We found that TBMS1 could dramatically increase the time of wound-healing in a dose-dependent manner (Figure 3a,b). To further confirm TBMS1 affects in glioblastoma cells migration and invasion, we performed Transwell assays. The results showed TBMS1 decreased the migration and invasion rates of U87 and LN229 cells (Figure 3c). Finally, Western blot assay was utilized to examine the expression of MMP-7, MMP-2, Snail and Slug, which play critical role in cell migration and invasion. We found TBMS1 could decreased the expression of MMP-7, MMP-2, Snail and Slug (Figure 3d). In conclusion, TBMS1 could reduce the expression of related migratory proteins in U87 and LN229 cells which cause the inhibition of glioblastoma metastasis.

### 3.4. TBMS1 Inhibits Glioblastoma Cells Activation of AKT, ERK Pathway by Enhancing MET Ubiquitination Degradation

In order to clarify the mechanism of TBMS1 inhibition of glioblastoma cells growth, migration and invasion. We checked the expression of several important protein for the growth of glioblastoma cells. The results showed the activation of MEK, ERK, and AKT are significantly inhibited (Figure 4a). The tyrosine kinase receptor plays a key role in the activation of downstream pathways ERK and AKT. Therefore, we guess whether TBMS1 has influence on RTK. In order to verify our conjecture, we detected the protein level of MET, which is an important oncogene in glioblastoma. We found TBMS1 highly reduced the expression of MET (Figure 4b). To further confirm our hypothesis, we constructed LN229 and U87 cell lines that overexpressed MET (Appendix A). We found cell lines which overexpress MET could significantly attenuate the inhibition of TBMS1 on growth, migration and invasion (Figure 4c and Appendix A) and the expression of p-ERK, p-AKT comparing to cell lines overexpressing vector (Figure 4d). Here, we found TBMS1 could decrease the express of p-AKT and p-ERK by down-regulating MET protein levels.

The reduction of MET protein levels mainly includes transcriptional and post-translational regulation. In order to further explore the mechanism, we examined the mRNA level of MET in the LN229 and U87 cell lines after dosing. There was no significant decrease in the mRNA level of the experimental group MET with the dose increasing (Appendix A). Therefore, we suspect the MET protein level decrease is due to changes in its level of ubiquitination. To verify our hypothesis, we firstly added MG-132, a proteasome inhibitor, to block the Ubiquitination degradation of MET. We found MG-132 could dramatically attenuated the decrease of MET caused by TBMS1 (Appendix A). Subsequently, we performed IP experiments on 293FT cells. The results demonstrated that TBMS1 significantly enhances the level of ubiquitination of MET (Figure 4e). Based on the above results, we found TBMS1 is able to inhibit the proliferation and migration of glioblastoma cells by enhancing the level of ubiquitination of oncogene MET.

### 3.5. TBMS1 Inhibits Glioblastoma Cells Clonogenicity, Tumorigenesis, and the Expression of MET

To detect the effects of TBMS1 in clonogenicity of glioblastoma cells, we performed soft agar assay. The results showed the number of cell clone formation decreased significantly with the dose increasing (Figure 5a). Moreover, in order to further explore the effects of TBMS1 on the tumorigenic ability of glioblastoma cells, we performed subcutaneous tumor experiment by U87 cells in mice. We found tumor weight and volume were significantly decrease in the TBMS1 group (Figure 5b–d). Subsequently, we examined the expression of MET and Ki-67 by immunohistochemistry. From the experimental results, we found that TBMS1 could significantly reduce the expression of MET and Ki-67 (Figure 5e). It is critical for a possible therapeutic application of TBMS1 against glioblastoma, whether TBMS1 cross the blood–brain barrier and accumulate into a tumor. Thus, we detected the influences of TBMS1 on intracranial tumor. The results showed TBMS1 could reduce the magnitude of intracranial tumor (Figure 5f,g). Together, TBMS1 have significant inhibitory effects on the clonal ability of glioblastoma cells and on the growth of subcutaneous and intracranial tumors.

## 4. Discussion and Conclusions

Chinese traditional medicine has a long history in the treatment of disease, and Rhizoma Bolbostemmatis is one of them. The highest-acquired active monomer TBMS1 which extracted from the tubers of Rhizoma Bolbostemmatis has proven its potential ability in the treatment of gastric cancer, liver cancer, cervical cancer, lung cancer, ovarian cancer, glioblastoma, esophageal cancer, etc [5,6,7,8,9,10]. At present, the anti-tumor mechanism of TBMS1 has been found to induce anti-tumor effects by inducing apoptosis [11,12], autophagy [13], cell cycle arrest [14,15], inhibition of cancer cell growth, metastasis [16], differentiation [17], angiogenesis [18] and inflammation [19]. Although the relevant critical pathways and molecular targets of TBMS1 have been confirmed, we need to further improve and explore the relevant mechanisms for the possible therapeutic application to the treatment of cancer. However, this paper found that TBMS1 can target its important oncogene MET in glioblastoma cells, providing a potential way for the treatment of gliobalstomatosis.

MET known as c-met, has the activity of a tyrosine kinase. It is usually express in epithelial cells and can participate in the regulation of cell proliferation, metastasis, secretion and other biological functions by the activation of ligand HGF (hepatocyte growth factor) [20]. Previous studies have found that oncogene MET has abnormal activation and gene amplification in a variety of tumors, such as glioblastoma or gastric cancer [21]. Due to its important functions, it has become a potential target for cancer treatment. There are many inhibitors for MET that have entered the clinical stage and achieved certain results. Therefore, the TBMS1 found in this paper could provide a possible way for targeted therapy of MET through ubiquitin degradation of MET. However, in order to further explore the mechanism of TBMS1 on the inhibition of proliferation and migration of glioblastoma. Validation of the relevant ubiquitin ligase to which TBMS1 acts is a direction of investigation. Moreover, the oncogene MET have abnormal activation and amplification in various tumor cells. Therefore, exploring whether TBMS1 have a relevant role in a variety of cells is still a puzzle that needs to be verified in the future.

Traditional Chinese medicine is a potential way for the treatment of cancer because of its well documented toxicity and reverse effects during its long course of application. In the last century, Rhizoma Bolbostemmatis was treated for gastric cancer, and has achieved good results. However, due to side effects such as nausea and vomiting [22], the utilization was abandoned. However, using the extracted active ingredients and taking a better method of administration, it is possible to eliminate the side effects while achieving the corresponding effects. From the above experimental results and analyses, we found TBMS1 could inhibit the proliferation and migration of glioblastoma cells by increasing the ubiquitination level of MET. It is a therapeutic potential Traditional Chinese medicine monomer target MET in glioblastoma cells.

## Figures and Tables

**Figure 1 cells-08-00774-f001:**
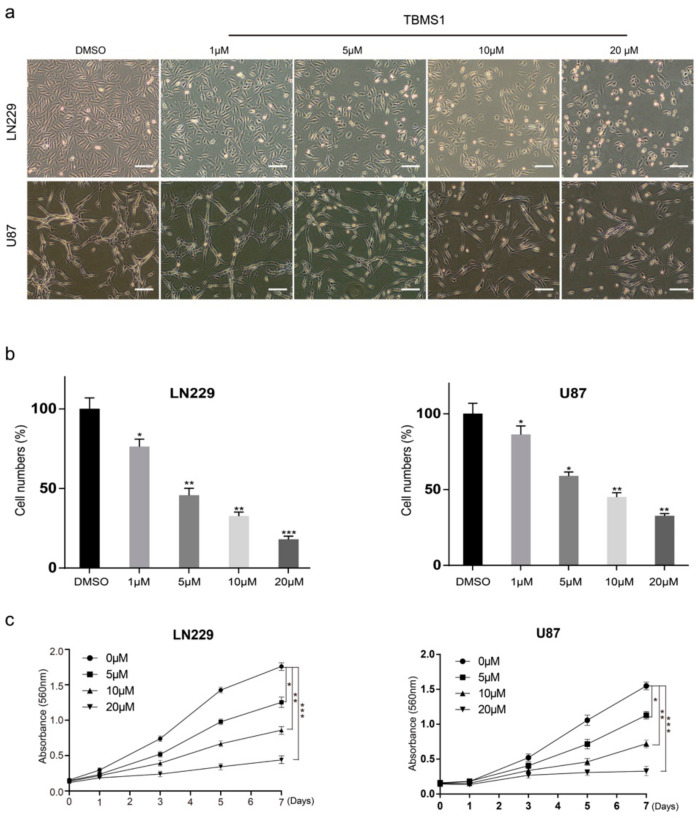
Tubeimoside-1 (TBMS1) inhibits cell proliferation in glioblastoma cells. (**a**) The phenotype of LN229 and U87 cells treated with different concentrations of TBMS1 for 24 h. DMSO was added as a control, and the scale was 200 μm. (**b**) Percentage of cells in each group, regard percentage of the cells in the control group as 100%. (**c**) Viability of glioblastoma cells treated with different concentrations of TBMS1. DMSO was added as a control. * *p* < 0.05, ** *p* < 0.01, *** *p* < 0.001, *p*-values < 0.05 were considered as statistically significant.

**Figure 2 cells-08-00774-f002:**
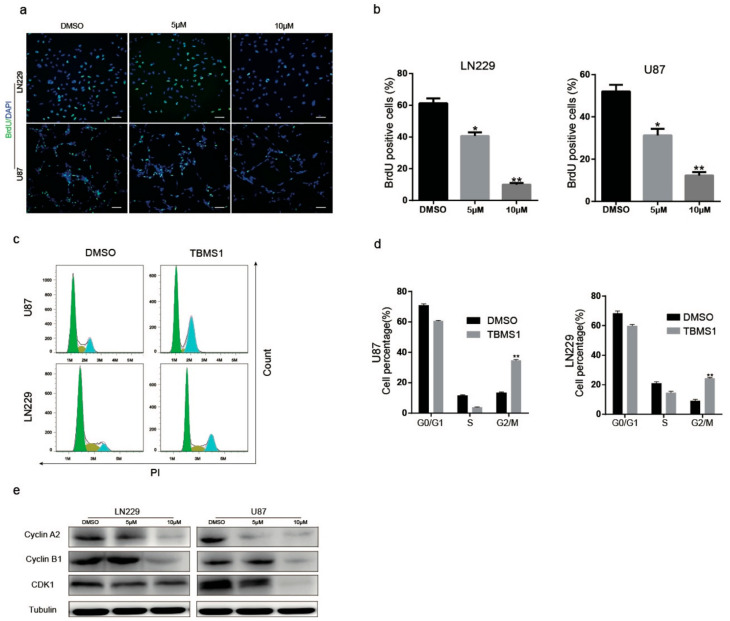
TBMS1 causes glioblastoma cells cell cycle arrest at G2 phase. (**a**) Changes in the number of BrdU-positive cells in LN229 and U87 cells treated with different concentrations of TBMS1 for 48 h. DMSO was added as a control, scale bar was 200 μm. (**b**) A statistical plot of the BrdU positive cell rate in panel (**a**). (**c**) Cell cycle detection of LN229 and U87 cells supplemented with DMSO or TMBS1 was performed by flow cytometry. (**d**) The graph of the percentage of the results of each cycle of the cells is statistically plotted in the graph (**c**). (**e**) The protein expression levels of CDK1, Cyclin A2, Cyclin B1 and Tubulin in LN229 and U87 cells treated with different concentrations of TBMS1 for 48 h. DMSO was added as a control. * *p* < 0.05, ** *p* < 0.01, *** *p* < 0.001, *p*-values < 0.05 were considered as statistically significant

**Figure 3 cells-08-00774-f003:**
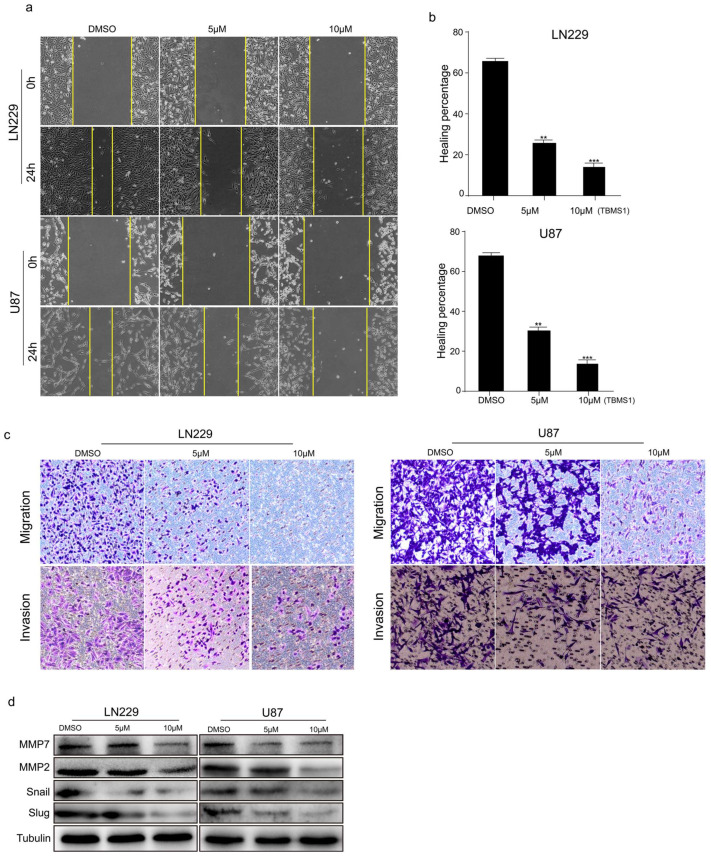
TBMS1 blocks glioblastoma cells migration and invasion. (**a**) The migration ability of glioblastoma cells treated with different concentrations of TBMS1 was measured by wound-healing assay. DMSO was added as a control. (**b**) According to (**a**) plot, the percentage of cell healing was counted under different TBMS1 concentrations. (**c**) The invasion ability of glioblastoma cells treated with different concentrations of TBMS1 was measured by Transwell assay. DMSO was added as a control. (**d**) The expression levels of MMP7, MMP2, Snail, Slug, and Tubulin proteins in LN229 and U87 cells were measured by western blot treated with different concentrations of TBMS1. DMSO was added as a control. ** *p* < 0.01, *** *p* < 0.001, *p*-values < 0.05 were considered as statistically significant.

**Figure 4 cells-08-00774-f004:**
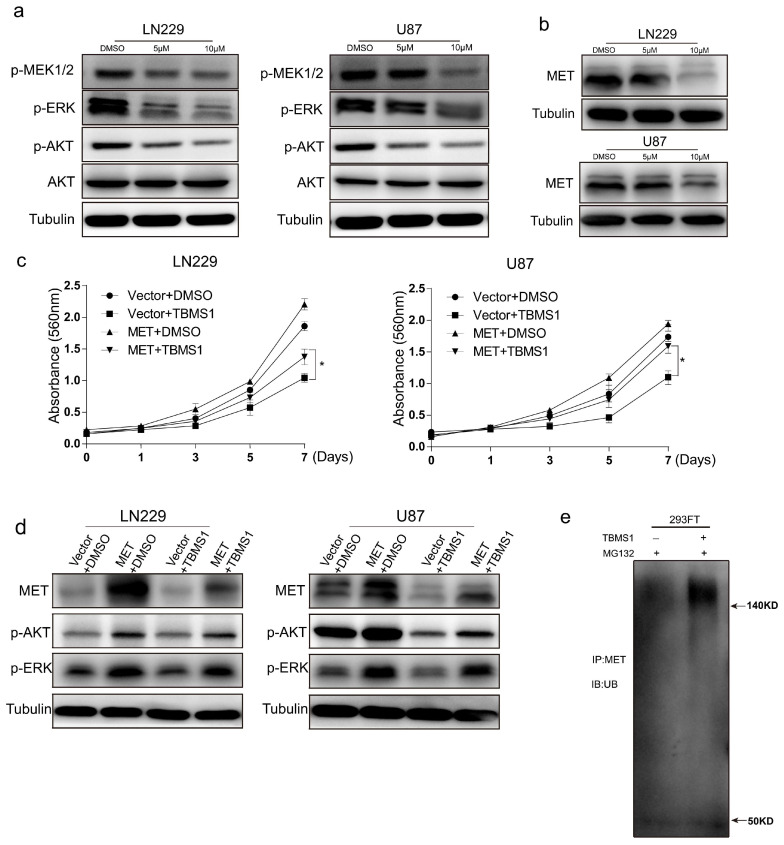
TBMS1 inhibits glioblastoma cells activation of AKT, ERK pathway by enhancing MET ubiquitination degradation. (**a**) The expression levels of p-MEK, p-ERK, p-AKT, AKT, and Tubulin proteins in LN229 and U87 cells were measured by western blot treated with different concentrations of TBMS1. DMSO was added as a control. (**b**) The expression levels of MET and Tubulin proteins in LN229 and U87 cells were measured by western blot treated with different concentrations of TBMS1. DMSO was added as a control. (**c**) Viability of LN229 and U87 cells overexpressing MET and vector under treatment with TBMS1 (5 μM), respectively. DMSO was added as a control. (**d**) The expression levels of MET, p-AKT, p-ERK and Tubulin proteins in LN229 and U87 cells overexpressed MET and vector respectively were measured by western blot. Cells were supplemented with TBMS1 (5 μM) and DMSO before the experiment. (**e**) The level of ubiquitination of the MET protein in 293FT cells after the addition of TBMS1 (5 μM) was examined using an IP assay. DMSO was added as a control. * *p* < 0.05, *p*-values < 0.05 were considered as statistically significant.

**Figure 5 cells-08-00774-f005:**
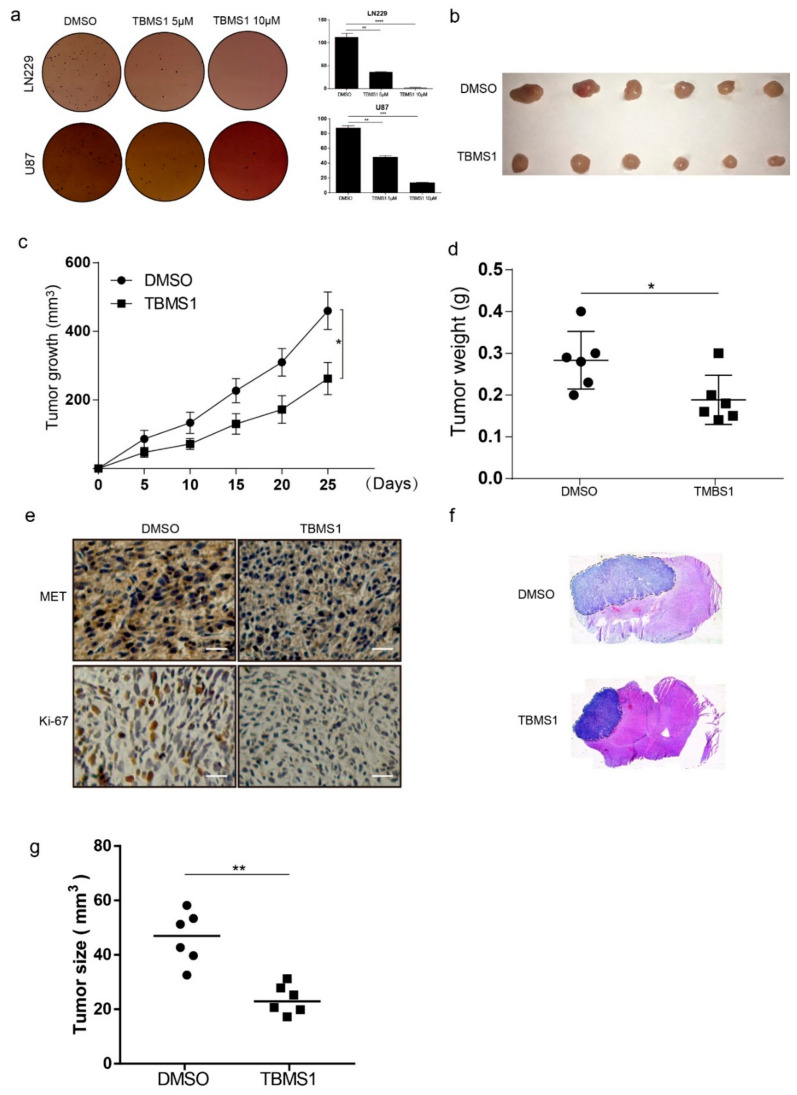
TBMS1 inhibits glioblastoma cells clonogenicity and tumorigenesis and the expression of MET. (**a**) The clonogenicity of LN229 and U87 treated different concentrations of TBMS1. On the left is the formation of clones after 3 weeks of culture, and on the right is a statistical plot of the number of clones formed. DMSO was added as a control. (**b**) After subcutaneous injection of U87 cells into mice, subcutaneous tumors obtained by injecting TBMS1 and DMSO for 25 days, respectively. (**c**) A graph of tumor growth in mice during dosing. (**d**) Subcutaneous tumor weight statistics after 25 days of mice administration. (**e**) Immunohistochemistry experiments were used to detect the expression levels of MET and Ki-67 in mouse tumor tissues. Scale bar = 200 μM. (**f**) After intracranial injection of U87 cells into mice, intracranial tumors obtained by injecting TBMS1 and DMSO for 23 days, respectively. (**g**) Size statistics of intracranial tumors. * *p* < 0.05, ** *p* < 0.01, *** *p* < 0.001, *p*-values < 0.05 were considered as statistically significant

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
