# Peer review of "Tubeimoside-1 Inhibits Glioblastoma Growth, Migration, and Invasion via Inducing Ubiquitylation of MET"

_cells, 2019, doi:10.3390/cells8080774_

Round 1

Reviewer 1 Report

Authors addressed my concern about ubiquitination using MG-132 inhibitor and showed normal-like cells (HEB) are more resistant to TBMS1 compared to tumor cells.

Minor comment: "Invasion" is misspelling on the supplementary material legend and figure 3.

In my opinion, the manuscript has been enhanced and deserves to be published.

Author Response

Response to Reviewer 1 Comments

Point 1: Minor comment: "Invasion" is misspelling on the supplementary material legend and figure 3.

Response 1: Thanks for your comment, I have corrected spelling mistakes.

Reviewer 2 Report

The intracranial injection experiment is unsatisfactory for the following reasons:

1)      No details on the experiments are provided

2)      How 10µl were injected? This experiment cannot be done without a peristaltic pump for which 2-5 µl are recommended. Anything above this volume will likely come back out of the brain (even using the pump), resulting in a tremendous variability in the number of tumor cells truly injected, assuming the mouse survives.  How many mice were used?

3)      The pictures of DMSO and the treated tumors in Fig 5f seem to have been taken from different angles (mid sagittal and coronal sections, respectively) making impossible to compare them. Tumor weight and size would have been more informative.

The manuscript still requires major editing for the English grammar.

Author Response

Response to Reviewer 2 Comments

Point 1: No details on the experiments are provided.

Response 1: We have provided the details on the experiments in 2.11 of manuscript.

Point 2: How 10µl were injected? This experiment cannot be done without a peristaltic pump for which 2-5 µl are recommended. Anything above this volume will likely come back out of the brain (even using the pump), resulting in a tremendous variability in the number of tumor cells truly injected, assuming the mouse survives.  How many mice were used?

Response 2: Firstly, I’m sorry for our mistake which gave you some misunderstand. In actually, 5ul with 1 × 105 U87 cells were injected into the brains of mice (the details were mentioned in 2.11). 6 mice were used for control and 6 mice were used for TBMS1 group.

Point 3: The pictures of DMSO and the treated tumors in Fig 5f seem to have been taken from different angles (mid sagittal and coronal sections, respectively) making impossible to compare them. Tumor weight and size would have been more informative.

Response 3: Because each tumor grows differently in the mice brain. In the HE staining experiment, we selected the mean size part of the tumor in the brain tissue. In spite of it from different angles, there are referable value. Tumor size have been provided in figure s3a.

Point 4: The manuscript still requires major editing for the English grammar.

Response 4: We will receive English editing service to checking grammar.

Round 2

Reviewer 2 Report

The author's response to my question for figure 5f is unacceptable. Images of HE for one control and one treated mouse taken from different angles are meaningless.Tumor size should be reported in the results sections and shown in the main text instead of supplemental figure. Again, this critical experiment requires more details and quantification of data. Intracranial injection has high variability. The authors need to show tumor size in every mouse from the experiment (12 mice total) as well as statistical analysis of the differences in tumor size between controls and treated animals. A plot graph with median would be more informative than a bar graph.

Author Response

Response to Reviewer 2 Comments

Point 1: Images of HE for one control and one treated mouse taken from different angles are meaningless.Tumor size should be reported in the results sections and shown in the main text instead of supplemental figure. Again, this critical experiment requires more details and quantification of data. Intracranial injection has high variability. The authors need to show tumor size in every mouse from the experiment (12 mice total) as well as statistical analysis of the differences in tumor size between controls and treated animals. A plot graph with median would be more informative than a bar graph.

Response 1: Firstly, thanks for your comments for our manuscript. We have changed figure 5f according to your advice and reported tumor size in the main text. We used a plot graph with median to show the tumor size of every mouse. Moreover, accurate value of 12 mice intracranial tumor are provided below. Thank you again for your advice. 

Tumor Size (mm3)
DMSO58.1832.5653.3642.72
DMSO39.6751.26

TBMS117.2319.8431.227.85
TBMS125.2120.65

This manuscript is a resubmission of an earlier submission. The following is a list of the peer review reports and author responses from that submission.

Round 1

Reviewer 1 Report

This study examines the anti-cancer properties of TBMS1 in glioblastoma cell lines U87MG and LN229. The authors found that TBMS1 treatment results in ubiquitination and degradation of the proto-oncogene MET, and that downregulation of MET is associated with decreased migration, proliferation and invasion of those cancer cells. There are many publications describing a variety of mechanisms involved in the effects of TBMS1 on tumor cell lines and the present study adds some new information. However, it could be improved by showing the effect on intracranial tumors rather than subcutaneous growth. In addition, Figure 4c shows a modest effect of MET on proliferation in the presence of TBMS1, arguing on the role of MET downregulation on the anti-cancer effects of this compound. Importantly, does TBMS1 cross the blood brain barrier? For a possible therapeutic application of TBMS1 against gliomas, it is critical to show efficacy of the compound in crossing the blood brain barrier and accumulate into a tumor.

Most of the results validate previous findings on the anti-cancer properties of TBMS1. Related to MET, although the authors show its importance in the TBMS1-mediated effects, they do not provide full information on the role of MET inhibition in the anti-cancer effects of TBMS1. Specifically, the authors should show the contribution of MET downregulation in migration and invasion, as they show for proliferation.

How TBMS1 was administered to the mice? Locally into the tumor or systemically?

Please revise the text for English grammar.

Reviewer 2 Report

Manuscript from Cao et al. describes the tubeimoside-1 effects on glioblastoma cell proliferation, apoptosis,  migration and invasion, tumor cell signaling, colony and tumor formation.  It’s an organized, well-illustrated and results in general are sound.

My only (and important) concern is about the ubiquitination result.  Fig 4E does not show clear MET degradation and in order to affirm that MET is decreased by degradation other experiments should be performed to support this conclusion. IP experiment should show the presence of MET and the Ubiquitin.

I’d suggest quantification of enzymes levels/activity involved in ubiquitination and/or deubiquitination and/or proteasome activity.  siRNA for ubiquitination proteins and treatment with tubeimoside-1 could also be helpful to prove this compound activity. Use of MG-132 to inhibit the proteasome as control.  
Also this study can be done using different ubiquitin mutants, mass spec, use of kits based on ubiquitin-specific affinity resins.
Positive and negative controls must be added.

Also, what are the effects of tubeimoside-1 on non-tumor cells? Is it tumor-specific? How’d be this effect on neurons? Is it possible to use this compound other than site-injection?  It should be discussed.

Minor: I believe the antibody company is Cell Signaling and not Cell Signal.